# Research Progress on Cordycepin Synthesis and Methods for Enhancement of Cordycepin Production in *Cordyceps militaris*

**DOI:** 10.3390/bioengineering9020069

**Published:** 2022-02-11

**Authors:** Li Wang, Huanhuan Yan, Bin Zeng, Zhihong Hu

**Affiliations:** 1Jiangxi Key Laboratory of Bioprocess Engineering, College of Life Science, Jiangxi Science & Technology Normal University, Nanchang 330013, China; wangli199711111@163.com (L.W.); huan89215@163.com (H.Y.); 2College of Pharmacy, Shenzhen Technology University, Shenzhen 518118, China

**Keywords:** *Cordyceps militaris*, cordycepin, biosynthesis, genetic modification

## Abstract

*C. militaris* is an insect-born fungus that belongs to *Ascomycota* and *Cordyceps*. It has a variety of biological activities that can be applied in medicine, health-care products, cosmeceuticals and other fields. Cordycepin (COR) is one of the major bioactive components identified from *C. militaris*. Thus, *C. militaris* and COR have attracted extensive attention. In this study, chemical synthetic methods and the biosynthesis pathway of COR were reviewed. As commercially COR was mainly isolated from *C. militaris* fermentation, the optimizations for liquid and solid fermentation and genetic modifications of *C. militaris* to increase COR content were also summarized. Moreover, the research progress of genetic modifications of *C. militaris* and methods for separation and purification COR were introduced. Finally, the existing problems and future research direction of *C. militaris* were discussed. This study provides a reference for the production of COR in the future.

## 1. Introduction

*C. militaris* is a traditional Chinese medicinal and edible fungus [1]. According to the phylogenetic classification system, *C. militaris* belongs to Ascomycota, Clavicipitaceae [2]. In recent years, multiple active ingredients and pharmacological properties of *C. militaris* have been studied [3]. *C. militaris* contains multiple biologically active substances such as COR, adenosine, cordyceps polysaccharide, cordyceps acid, fatty acids, amino acids, vitamins, ergosterol and myriocin [4,5,6,7]. Among these compounds, COR is regarded as one of the vital active secondary metabolite components in *C. militaris* [8,9,10]. COR was first separated from the culture filtrate of *C. militaris* in 1950 [11], and its structure was identified as 3’-deoxyadenosine in 1964 [12]. Except for *C. militaris*, COR can also be produced by *Aspergillus nidulans* (*A. nidulans*) [13], *Cordyceps kyushensis Kob* (*C. kyushuensis Kob*) [14] and *Cordyceps cicadae* (*C. cicadae*) [15].

As COR is a nucleoside analog compound and a member of the adenosines, it is recognized as a biologically active metabolite with therapeutic potential [16,17]. An increasing number of studies have demonstrated that COR exhibits an extensive range of biology activities [18,19,20]. COR inhibited tumor growth and elongated survival time of tumor-bearing mice by regulating the expression of cytokines such as IL-2, IL-4, TNF-α, TGF-β and IFN-γ [21]. It exhibits hypoglycemic activity in liver of alloxan-induced diabetic mice by regulating glucose metabolism [22,23] and reduces the weight of high-fat diet-induced obese rats through regulating gut microbiota [24]. COR also has function of antiviral, anti-bacterial and anti-fungal activities. It shows antiviral effects to a number of viruses including influenza virus [25], plant viruses [26], HIV [27], murine leukemia virus [28], Epstein–Barr virus (EBV) [29] and herpes [30]. Studies revealed that COR suppresses the transfer of EBV to AGS cells [31] and selectively inhibits influenza viral genome replication [32]. COR can effectively suppress the growth of both Gram-positive and Gram-negative bacterial pathogens by damaging cell membranes and genomic DNA [33]. It also exhibited potent antifungal activity as treatment of COR prolonged the survival time and decreased CFU in the kidneys of a candidiasis invasive murine model [34]. COR inhibits human platelet aggregation in a cyclic AMP- and cyclic GMP-dependent manner [35]. In addition, COR can also protect against inflammatory injury of many diseases including acute lung injury [36], asthma [37], rheumatoid arthritis, Parkinson’s disease, hepatitis, atherosclerosis, and atopic dermatitis by regulating the NF-κB, RIP2/Caspase-1, Akt/GSK-3β/p70S6K, TGF-β/Smads, and Nrf2/HO-1 signaling pathways [38]. Furthermore, COR inhibits ultraviolet B (UVB)-induced matrix metalloproteinase expression [39] and the activities of tyrosinase, tyrosinase-related protein-1 (TRP1) and tyrosinase-related protein-2 (TRP2) [40], indicating it can be used in cosmetics for photoaging prevention and depigmenting of skin. Besides, COR is also a plant growth inhibitor that can be potentially used as novel alternative to glyphosate [5]. In summary, COR showed a lot of biological activities including anti-cancer, anti-tumor, anti-diabetic, anti-obesity, anti-herpes, anti-bacterial, anti-fungal, anti-transfer, anti-platelet aggregation, anti-inflammatory, anti-photoaging, anti-pigmentation, and plant growth inhibitor. As many of these biological activities are revealed in cells, further studies should be performed to illustrate the molecular mechanism and activities in animals. Because natural drugs are deemed to be safer and more valid with fewer side effects, COR can be used as a functional component in health care products, foods, and cosmetics [41,42]. Therefore, medicinal *C. militaris* may be one of the most important valuable herbal medicines.

In the traditional procedures, COR is mainly obtained from natural *C. militaris* fruiting body [43]. However, the amount and extraction efficiency from natural *C. militaris* fruiting body can not satisfy market needs because this process showed long production period, low yield, high running cost, high energy consumption and low biotransformation ration [44]. Because of the scarcity and value of this wild resource, it cannot meet market demands [45,46]. Therefore, many studies have focused on the synthesis of COR and improving the content of COR in *C. militaris*. The identification of the COR chemical formula makes chemical synthesis a possible way to produce it. However, the chemical synthesis showed high cost and low yield. On the one hand, this review summarizes the chemical and biosynthetic pathways of COR and compares their advantages and disadvantages. On the other hand, the methods of improving COR content by optimizing the conditions of liquid and solid fermentation are summarized. The research progress of genetic modifications and the improvements of COR content by molecular biology methods in *C. militaris* are also reviewed [47]. In addition, the separation and purification methods of COR are introduced. Therefore, this study aims to provide references for COR production and improvement of the content of COR.

## 2. Chemical Synthesis of Cordycepin (COR)

The chemical synthesis methods of COR include semi-synthesis and total synthesis routes. Semi-synthesis refers to the use nucleoside derivatives as substrate to produce COR. Semi-synthesis of COR was first reported in 1960 [48]. In this method, 3’-O-p-nitrobenzenesulfonyl adenosine was used as the starting substrate and reacted with anhydrous sodium iodide in hot acetonylacetone to produce 3’-deoxy-3’-iodoadenosine, which was then converted into 3’-deoxyadenosine by catalytic hydrogenation. All materials in this route were commercially available and the synthetic procedure was simple. However, because of poor selectivity of 2’-hydroxyl group and 3’-hydroxyl groups, 5’-O-acetyl adenosine can be obtained with two isomers after acylation. The yield of 3’-acylation product is low, and the overall yield of the last reaction step is only 17%, resulting in the overall yield of this route is less than 14%. The difference between adenosine and COR is only one hydroxyl group. Thus, methods to synthesis COR by using adenosine as substrate were developed. Hansske et al. synthesized COR by treating 2’,3’-anhydroadenosine with lithium triethyl borohydride (or deuteride). This is a high yield three-stage synthesis of COR from adenosine derivative and the overall yield reached 90% with no 2′-deoxy isomer detected, but the substrates were expensive [49]. Another method is to directly use adenosine as substrate to synthesis 3’-deoxyadenosine by phenyl chlorothiocarate [50,51]. The product yield of this route is relatively high (the overall yield of four steps is 56%, and each step is about 90%). However, Bu_3_SnH is involved in the reaction, and it may lead to high content of tin residual in the final product of COR. In 2000, Aman et al. used 2-acetoxyisobutyryl bromide (AIB, Mattock’s bromide) to convert natural adenosine to the corresponding adenosine bromoacetate, followed by acid hydrolysis and hydrogenolysis to obtain COR [52]. The reaction conditions of this method are mild and controllable, and the purity of final COR reached 99%. However, the overall yield of the route is about only 20% because of the wastage in the reduction reaction and crystallization purification process. Besides, the industrial production cost of this method is relatively high. Huang et al. designed two independent methods to synthesize COR by using adenosine as substrate [53]. The one is to convert the 3′-OH of adenosine into an iodide group and further dehalogenation to produce COR. Although this method presented a short synthetic procedure, the overall yield was only 13.5%. In order to improve the overall yield of COR, another synthetic method was studied, and this route was consisted of four individual steps: (1) 5′-OH in adenosine was protected by triphenylmethyl chloride (TrCl), (2) the 3′-OH was reacted with p-TsCl, (3) a reducing agent used to remove the -O-tosyl (-OTs) group, (4) acetic acid and dichloromethane solution were used to remove the triphenylmethyl protected group to obtain COR. By this way, the overall product yield of COR reached 36%, 2.6 times higher than their former method. The complete synthesis route of COR was first reported to use a non-nucleoside derivative, di-hydrofuranmethanol, as substrate [54]. This is asymmetric syntheses of COR by cycloisomerization of alkynyl alcohols to endocyclic enol ethers. The chemical reagents used in this route are expensive, and the overall yield of this method is about only 14%. Another complete synthesis route using D-glucose or D-xylose as raw materials was also developed [55]. The key steps of this method are the Barton–McCombie reaction to remove the 3′-hydroxyl group of glucose and xylose to synthesize 3′-deoxyribose derivative. The total yield of the two routes were 37% and 40% respectively, and the product purity reached 98.5%. The raw materials in this method are cheap and available, and this is a very potential method for the large-scale production of COR after optimization. The chemical synthesis methods of COR are shown in Table 1.

## 3. Biosynthesis of COR

As described, COR is 3’-deoxyadenosine. Thus, adenine and ribose were proposed as a potential precursor of COR [56,57]. In 1976, [U-14C] adenosine and [3-3H] ribose was used to study the biosynthesis pathway of COR in *C. militaris*. By this way, it suggested that COR was directly biosynthesized by converting adenosine to 3’-deoxyadenosine without hydrolysis of the N-riboside bond, similar to that for the formation of 2’-deoxynucleotides, an intermediate required for nucleotides metabolism [58]. Further studies showed the accumulation of adenine or adenosine leads to the increase of COR in *C. militaris*, indicating adenine or adenosine may be the precursor of COR [59]. However, due to the lack of genome sequence information, its biosynthesis pathway has not been elucidated for a long time. It was not until 2011 that the whole genome sequencing of *C. militaris* was completed, which revealed the genetic and metabolic mechanism of COR [60]. According to the KEGG annotation of *C. militaris*, Zheng et al. identified that *C. militaris* possesses most of the genes required for metabolism of adenine and adenosine and constructed the biosynthesis route of deoxyadenosine based on the existed enzymes involved in purine and adenosine metabolic pathways (Figure 1). In this model, 5’-nucleotidase was an important enzyme involved in COR biosynthesis. However, another two important enzymes ribonucleotide trisphosphate reductase (RNR; converts ATP to dATP) and deoxyadenosine kinase (DAK, converts deoxyadenosine to dAMP) that required in this proposed pathway were lacked in *C. militaris* genome. In 2014, a similar putative biosynthesis pathway for COR was also proposed in *O. sinensis* [61]. In this study, all the enzymes required for COR biosynthesis were identified by GO and KEGG analysis. It also suggested that NT5E is a key enzyme for the production 3′-deoxyadenosine from 3′-dAMP, and revealed ADK, ADEK, and NT5E, which are involved in phosphorylation and dephosphorylation in adenosine metabolic pathway, may also be involved in phosphorylation and dephosphorylation in the COR biosynthesis pathway. In 2016, Lin et al. predicted and verified a similar biosynthetic pathway of COR in *Hirsutella sinensis* (*H. sinensis*) [62]. The core of these proposed COR biosynthetic pathways can be summarized as follows: AMP was converted to ADP catalyzed by ADEK and ADP was converted to 3′-dADP (3′-deoxyadenosine 5′-diphosphate) catalyzed by ribonucleotide reductase (NRDJ), then 3′-dADP was converted to 3′-deoxyadenosine 5′-phosphate (3′-dAMP) catalyzed by ADEK and finally 3′-dAMP was converted to COR by NT5E (Figure 1). However, this proposed COR biosynthetic pathway was mainly based on bioinformatics analysis, lacking genetic or molecular biology evidence.

In order to investigate the COR biosynthesis pathway in *C. militaris*, Xia et al. performed gene knock out to examine and verify that the three genes coding for enzymes involved in the conventional nucleoside/nucleotide metabolic pathways, adenine phosphoribosyltransferase (cmAPRT), 5′-nucleotidase (cmNT5E), and nucleoside triphosphate pyrophosphatase (cmNTP), are not responsible for COR production in *C. militaris* [63]. Thus, Xia et al. performed comparative genomic research between *A. nidulans* and *C. militaris* as *A. nidulans* can also produce COR [13,63]. Through biological information and gene function research, this clarified a gene cluster responsible for COR biosynthesis. The gene cluster was consisted of four physically linked and highly conserved genes, designated as CCM-04436-CCM-04439 (Figure 2A), named Cns1-Cns4. The proteins encoded by these four genes have different conserved domains (Figure 2B). Cns1 contains oxidoreductase/dehydrogenase domain, Cns2 contains metal-dependent phosphohydrolase domain and belongs to HDc family protein, Cns3 contains a nucleoside/nucleotide kinase (NK) at N-terminal and HisG domain at C-terminal, Cns4 is a putative ATP-binding cassette type of transporter (ABC transporter). By gene knockout validation, it proved that Cns1 and Cns2 are indispensable for COR biosynthesis, Cns3 is responsible for pentostatin (PTN, 2′-deoxycoformycin, another adenosine analog) production. PTN production can inhibit the activity of ADA to prevent COR deamination to 3′-dI (Figure 2C). Cns4 contributes to the conjugative biosynthesis of PTN to protect COR from deamination and is a self-detoxification mechanism of COR used to regulate excessive production of COR (Figure 2C). The biosynthetic pathway for COR in *C. militaris* is revealed as follows: Cns3 catalyzed hydroxyl phosphorylation of the 3′-OH position on adenosine to produce 3′-AMP (adenosine-3′-monophosphate); then, 3′-AMP is dephosphorylated to 2′-carbonyl-3′-deoxyadenosine (2′-C-3′-dA) catalyzed by Cns2, and finally, 2′-C-3′-dA was converted to COR catalyzed by the oxidoreductase Cns1. This study also revealed that Cns1 and Cns2 interact tightly with each other and one enzyme cannot function without the other. Consistent with the results of Xia et al., similar biosynthetic pathways of COR have been found in *C. kyushuensis Kob* [14] and *C. cicadae* [15]. By transcriptome and proteomics analysis of *C. kyushuensis Kob*, it identified four related genes named ck1-ck4 are response for COR synthesis. The function of ck1-ck4 was similar with Cns1-Cns4 originated from *C. militaris*. Recently, Wongsa et al. proposed alternative pathways for COR biosynthesis by comparing the transcriptome profiles of *C. militaris* cultured in xylose, glucose and sucrose, and identified some genes required in this pathway [64]. In accordance with the study by Xia et al. [63], the COR biosynthesis pathway is highly correlated with the formation of the precursor 3′-AMP, which can be synthesized by 2′,3′-cAMP, a byproduct of mRNA degradation [65]. Two genes (Unigene8329 and Unigene6827) encoding 2′,3′-cyclic-nucleotide 2′-phosphodiesterase catalyzed the transformation of 2′,3′-cAMP to 3-′AMP. The study also proved the 2′-C-3′-dA derived from 3′-AMP is an intermediate precursor of COR and Unigene5711 encodes a redox domain protein that converts 2′-C-3′-dA into COR [66].

## 4. Liquid Fermentation to Produce COR

Currently, commercially used COR was mainly isolated from *C. militaris* fermentation [67]. As COR content are very different in strains from different places, screening strains with high COR content is the first issue to be considered in COR production. In order to improve the COR content, many optimization methods were studied. For liquid fermentation, the optimization measurements main include the following aspects: screening strains with high COR content, ion beam irradiation, adding agents into the medium, medium composition optimization, two-stage shaking-static culture and LED irradiation. A high COR producing *C. militaris* strain named KSP8 was screened through mating-based sexual reproduction [68]. Due to genetic recombination occurring during the sexual reproduction, COR content in KSP8 increased 116–164% comparing with average content of COR in 12 parent strains (3.06 g/L in brown rice medium). The KSP8 strain was inoculated in brown rice (mixing 40 g brown rice and 4 g silk worm pupae in 64 mL liquid medium) and silk worm pupae (20 g brown rice, 40 g silk worm pupae in 64 mL liquid medium) mediums, respectively; the liquid medium formula was as follows: 20 g sucrose, 20 g peptone, 1 g MgSO_4_·7H_2_O, 0.5 g KH_2_PO_4_, and 1 L distilled water. By using ion beam irradiation, a *C. militaris* mutant strain G81-3 was obtained, and the yield of COR reached 6.84 g/L in surface liquid culture (the optimal concentrations of glucose and yeast extract for the mutant strain were 86.2 g/L and 93.8 g/L, respectively), which was 2.79 times than that of the corresponding wild strain (2.45 g/L) [69]. As adenosine was supposed to the substrate of COR, the addition of adenosine can also increase COR content. The yield of COR increased to 8.60 g/L by using G81-3 strain with 6 g/L adenosine addition [70], and this is the highest yield reported so far. Attempts of adding other various compounds such as purine biosynthesis-related compounds, coenzymes and surfactants to the basal medium for the purpose of enhancing the COR production have tried. The results showed L-glycine, L-alanine, L-aspartic acid, L-glutamine and adenine were effective additives and the optimal additive combination was: 1 g/L adenine and 16 g/L glycine, resulting in 2.50 g/L COR, which was 4.1 times than that in the basal medium [59]. Carbon and nitrogen sources play important roles in liquid fermentation medium because these nutrients are directly related to cell proliferation and metabolite biosynthesis, and the optimized carbon and nitrogen sources for COR production was 42.0 g/L glucose and 15.8 g/L peptone, the content reached to 0.35 g/L, increasing 40% compared with control [71]. A two-stage fermentation process according to the Box–Behnken experimental design and response surface analysis was developed: combination 8 d of shake-flask fermentation with 16 d of static culture at pH 6 with 45 g/L yeast extract. By using this two-stage fermentation, the maximum production of COR was 2.21 g/L, which was two times than that of control (1.04 g/L) [72]. Studies also showed that Mg^2+^, Na^+^, Ca^2+^, Fe^2+^ and NO^3−^ are beneficial to the accumulation of COR [73]. For example, Fan et al. found supplement of 1 g/L of FeSO_4_ in the medium, increased 70% COR compared with control, and the content reached 0.60 g/L [74]. Researchers also found treatment of *C. militaris* with porcine liver extracts (10 g/L) can improve COR production in surface liquid medium, and the maximum content of COR reached 2.45 g/L, 4.9 times than that of control (approximately 0.5 g/L). Meanwhile, porcine liver extracts treatment combined with blue light (LED 440–450 nm) irradiation (16 h/d) can further promote the accumulation of COR to 3.48 g/L [8]. Cai et al. isolated a *Paecilomyces hepiali* (*P. hepiali*) OR-1 strain that can produce COR from plateau soil [75]. Subsequently, ^60^Co γ-ray and ultraviolet irradiation were employed to increase the COR content, and resulted in a high-yield mutant strain *P. hepiali* ZJB18001 with the COR content of 0.61 g/L, 2.3-fold that of the wild-type strain (0.26 g/L), screened. Furthermore, medium composition optimization (peanut oil 21 g/L, yeast extract 9.8 g/L, Fe_2_(SO_4_)_3_ 4.68 mmol/L and adenosine 2.14 g/L) based on Box–Behnken design and response surface methodology facilitated the enhancement of COR yield to 0.96 g/L at 25 °C for five days in submerged cultivation. The optimizations and contents of COR for liquid fermentation are shown in Table 2. Currently, liquid fermentation has become an important means of production of COR. Scholars have carried out a lot of research on optimization of medium and process conditions, and established a series of process engineering strategies to improve the production of COR.

## 5. Solid-State Fermentation to Produce COR

The production cycle of COR by solid-state fermentation (SSF) is relatively longer than that of liquid fermentation, which takes about 30 days. However, the fruiting body can be obtained by SSF. Thus, by contrast with liquid fermentation, COR content produced from SSF main detected in the fruiting body. To improve COR content, different optimizations have implemented. The strategies include, UVB and LED (different wavelength) radiation, pH, temperature, culture time and solid substrate composition optimizations. It has been reported that UVB light irradiated dry samples (buckwheat and embryo rice contained COR 0.08 mg/g) fermented by *C. militaris* was 1.1-fold higher than that of the irradiated fresh samples fermented (0.07 mg/g) [76]. Adnan et al. focused on effect of fermentation conditions such as pH, temperature, incubation time and solid substrates (wheat, oat and rice) on the production of COR and revealed the best possible combination for maximum COR production of temperature, pH and incubation time was: 25 °C, 5.5 and 21 days. Among the solid substrates, COR showed the highest production in rice medium (814.60 mg/g) followed by oat and wheat medium (638.85 and 565.20 mg/g, respectively) [77]. Wen et al. studied the optimization of solid-state fermentation for fruiting body growth and COR production by *C. militaris* and found the optimal culture substrate was brown rice [78]. The optimal fruiting body growth and COR production were observed at relatively low pH value. The optimal medium composition for fruiting body growth and COR production were as follows: 40 g/L glucose, 5 g/L peptone, 1.5 g/L MgSO_4_·7H_2_O, 1.5 g/L K_2_HPO_4_, 1.0 mg/L NAA, and 10 g/L glucose, 10 g/L peptone, 1.0 g/L MgSO_4_·7H_2_O, 1.0 g/L K_2_HPO_4_ and 1.0 mg/L NAA. These optimization strategies in solid medium culture led to a 68% (1.73 g/bottle) increase in fruiting body yield and a 63% (9.17 mg/g) increase of COR yield in fruiting body. Light is an important factor in the production of fruiting bodies and bioactive compounds in *C. militaris* [79]. Chiang et al. studied the effects of LED wavelengths on the production of bioactive compounds in *C. militaris* cultivated on brown rice [80]. The finding suggested that the optimal illumination times for COR production (3.97 mg/g) were 12 h/day, by fluorescent lamps and the favorable effect of wavelength for COR (2.89 mg/g) production was green light (526-531 nm). Among the 10 LED wavelength combinations, 3R:3B (the ratio of red light to blue light) was the most effective ratio of LED for COR production, which can increase the content of COR to 30 mg/g. The optimizations and contents of COR for liquid fermentation are shown in Table 3.

## 6. Genetic Modification of *C. militaris*

Genetic engineering refers to overexpression, knock down or gene editing by transgenetic approaches. It is an effective method to improve the content of bioactive components and a potential strategy for enhancement of COR synthesis in *C. militaris* [81]. The transformation methods commonly used in *C. militaris* include *Agrobacterium tumefaciens*-mediated transformation (ATMT), protoplast-mediated transformation (PMT), particle bombardment (PB), split marker approach and CRISPR-Cas9 system. In 2011, Zheng et al. constructed ATMT in *C. militaris* and used this method to construct a mutant library [82]. In this way, genes involved in degeneration during fruiting body production were successfully identified. In filamentous fungi, the *laeA* gene is regarded as a global regulator of secondary metabolism. Rachmawati et al. established the transformation system of *C. militaris* by using benomyl and its resistance gene as a marker system, and they transferred the *laeA* gene (under the control of *Beauveria bassiana Pgpd* promoter) into the *C. militaris* genome through PMT in 2013 [83]. The transformants harboring heterogeneous *laeA* showed enhanced production of secondary metabolites, compared to the wild-type strains. In 2015, Mao et al. transformed and expressed green fluorescent protein (GFP) in *C. militaris* by PB transformation [84]. In 2018, a split-marker approach based genetic transformation system was developed in *C. militaris*. This study constructed the linear and split-marker deletion cassettes and introduced them into *C. militaris* through PMT to knockout out a gene encoding a terpenoid synthase (Tns) [85]. The transformation of split-marker fragments showed higher positive efficiency (4.53 cfu/µg, 13.24%) of targeted gene disruption than the transformation of linear deletion cassettes (0.73 cfu/µg, 1.36%). In 2020, Lou et al. successfully used the split-marker method to knock out *Cmfhp* gene in *C. militaris* and revealed that the *Cmfhp* gene regulated NO and carotenoids biosynthesis, fruiting body development, and conidia formation [86]. This method proved to be an effective way of targeted gene deletion and it can be applied to study COR synthesis genes in *C. militaris*. In 2018, CRISPR-Cas9 system was applied in *C. militaris* by using codon-optimized cas9 under control of Pcmlsm3 promoter and Tcmura3 terminator for the first time [87]. The establishment of an efficient CRISPR/Cas9 gene destruction system (1.7 cfu/µg, positive efficiency is 64.7%) in *C. militaris* can greatly improve the genome reconstruction of *C. militaris*, and the biosynthetic pathway of COR will be the be edited to increase its content. The transformation methods for *C. militaris* are shown in Table 4 and Figure 3.

## 7. Genetic Modification of *C. militaris* to Increase COR Content

Recently, the content of COR was improved of by genetic modification of COR biosynthesis related genes. Light is an important factor in regulating pigment and fruiting body formation of *C. militaris* [88]. Wang et al. screened a spontaneous albino mutant strain (505 CGMCC 5.2191, 6.700 mg/g) of *C. militaris* with photoreceptors gene mutation and found carotenoid content was significantly decreased, while the COR was accumulated compared with its normal sibling strain (498 CGMCC 5.2190, 3.090 mg/g) [89]. Transcriptome analysis revealed the expression of pyruvate kinase (CCM_05734, 07110), and adenylate kinase (CCM_02335), adenosine nucleosidase (CCM_09682), and adenine deaminase (CCM_07169) were directly or indirectly regulated by blue-light receptor CmWC-1 [90]. The deletion of *CmWC-1* caused albino and dereliction of fruit body development, leading to a significant reduction in COR production (0.005 mg/g) and the *CmWC-1c* complemented strains restored the carotenoids and COR production (0.012 mg/g). Zhang et al. studied the role of ribonucleotide reductases (RNRs, the two RNR subunits, RNRL and RNRM) in the biosynthesis of COR by over expressing RNRs genes in *C. militaris* [91]. The results showed that the COR was significantly higher in RNRM overexpressed *C. militaris* (3.750 mg/g), whereas in the RNRL overexpression strain (2.600 mg/g) the COR content was not remarkably changed compared with the wild type (2.500 mg/g). It was speculated that RNRM can probably directly participate in COR biosynthesis by hydrolyzing adenosine. However, the mechanism of RNRM in COR synthesis needs to be further explored. A COR high-producing strain ZA10-C4 was obtained by screening the selectable marker MAT gene and biological characteristics [92]. The hybrid strain ZA10-C4 showed higher content of COR (357.166 mg/g) compared with the two parental strains (170.86 and 261.16 mg/g, respectively). L-alanine addition could significantly improve COR production. Transcriptome analysis of *C. militaris* with doubled COR production induced by L-alanine addition found two key Zn2Cys6-type transcription factors CmTf1 and CmTf2 played verified roles in promoting COR production, and overexpression of these two transcription factors also resulted in doubling the COR content of *C. militaris* [93]. In recent years, molecular biology methods have been used to study *C. militaris*, and the content of COR has been changed through genetic modification. Besides, overexpression of COR biosynthetic gene *Cns1* and *Cns2* can also produce COR in *S. cerevisiae* [63]. Although the content of heterologous biosynthesis COR in *S. cerevisiae* is very low, this is a potential way of producing COR. With the illumination of biosynthesis and regulation mechanism of COR, genetic modification of COR biosynthesis related genes in *C. militaris* will be a feasible way to increase the yield of bioactive substances (such as COR) in *C. militaris* [81]. The changes of COR content in different *C. militaris* strains by genetic modification are shown in Table 5.

## 8. Separation and Purification of COR

As COR is harmful to cell survival, the biosynthesized COR either secrete to the medium or are stored in organelles. Thus, according to the different sources of COR, the separation and purification methods are different. For solid fermentation, COR should be separated from both fruiting body and medium; for the liquid fermentation, COR can be separated from fermentation broth as most COR was secreted into the medium. Many methods for separation and purification of COR have been reported, including: supercritical fluid extraction (SFE), high-speed counter-current chromatography (HSCCC), macroporous resin adsorption, column chromatography and molecular imprinting.

Ling el al. used SFE and HSCCC to extract and purify COR from *C. kyushuensis* and the optimal SFE conditions is: 40 MPa, 40 °C, methanol (0.04 mL min^−1^) with a flow rate of CO_2_ (2.0 L min^−1^) [94]. The collected fractions were then analyzed by high-performance liquid chromatography (HPLC) to identified 8.94% COR in crude extract. Then, 98.5% purity of COR was obtained by one-step HSCCC with a two-phase solvent system composed of ethyl acetate-n-butyl alcohol-water at an optimized volume ratio of 1:4:5. HSCCC technique in a preparative scale also has been applied to separate and purify COR from the extract of *C. militaris* (Link): By one-step separation, a high efficiency of HSCCC separation was achieved on a two-phase solvent system of n-hexane-n-butanol-methanol-water (23:80:30:155, *v*/*v*/*v*/*v*), resulting in 64.8 mg COR with a purity of 98.9% from 216.2 mg crude sample (732 cation-exchange resin clean-up), and the rate of recovery was 91.7% [95]. Macroporous resin NKA-II adsorption was used to separate COR from the fruiting body of *C. militaris* [96]. In this way, COR in the crude sample prepared was 3.4%. The crude sample was further purified by recycling HSCCC with ethyl acetate, n-butanol, 1.5% aqueous ammonium hydroxide (1:4:5, *v*/*v*/*v*) as the optimized two-phase solvent system. Finally, 15.6 mg (98.5% purity) COR was obtained from 500 mg of crude sample. Zhang et al. purified COR from fermentation broth of *C. militaris* by using macroporous resin AB-8 and octadecyl-bonded silica chromatography [97]. The elution sample obtained was then loaded onto reversed-phase (RPC) with octadecyl bonded silica (ODS) as stationary phase and ethanol (95%, pH6.0)-acetic acid solution (pH6.0) as mobile phase. Then, the COR was isolated by crystallization and recrystallized, and the final purity of COR reached 99%. Surface imprinting technology was also used for COR purification [98]. In this method, a surface molecular-imprinted polymer for COR was synthesized. According to the adsorption kinetic curve, molecularly imprinted polymers (MIPs) obtained the maximum adsorption amount at 95.4 mg/g and 98% purity COR was obtained with the recovery of 25.67%. The methods for COR separation and purification are shown in Table 6 and Figure 4. It is difficult and tedious to separate and purify COR with a single stationary phase. Therefore, in order to further enhance the quality, efficiency and high recovery rate of separation and purification, the separation process conditions of COR should be systematically optimized in the future. Usually, many methods were combined to form a high efficiency and low-cost separation process.

## 9. Conclusions and Discussion

COR has gained much attention due to its various clinical functions and the market demand calling for the development of efficient production technology. Although different chemical synthesis methods of COR have developed, many problems still existed, including: a complicated synthesis process, cumbersome purification process, harmful organic solvents used in the synthesis process, and high cost. Therefore, chemical synthesis of COR is not suitable for large-scale industrial production at present. It takes a long time to clarify the COR biosynthetic pathway due to lack of genomic sequences of *C. militaris*. Two models of the COR biosynthetic pathway were proposed. The one was constructed based on genomic and transcriptome information analysis of genes involved in traditional adenine and adenosine metabolism in *C. militaris*, *O. sinensis* and *H. sinensis*. This model was similar to the metabolic pathway of nucleotides metabolic intermediate, 2’-deoxynucleotides. However, the genetic and molecular biology evidences are very limited. Besides, it has finally been proved that *O. sinensis* cannot produce COR [63]. The other pathway was also revealed by comparative genomics analysis and proved by genetic and molecular experiment. In this model, COR biosynthesis genes were located in a gene cluster consisting of four genes named, Cns1-Cns4. And this pathway was also proved in other COR producing fungi such as *C. kyushuensis Kob* and *C. cicadae*. However, the biochemical function and catalytic mechanism of the enzymes required in the pathway still needs to be further revealed. Further studies should be performed to prove whether both pathways existed in other fungi.

Nowadays, commercially used COR is mainly isolated from *C. militaris* fermentation. Liquid fermentation and solid culture have been used for COR production. The fermentation condition of liquid fermentation is easy to control, while the fruit body can be produced by solid culture. Currently, liquid fermentation has become an important means of production of COR. Scholars have carried out a lot of research on optimization of medium and process conditions, and established a series of process engineering strategies to improve the production of COR. Although the fruiting body of *C. militaris* can be artificially cultivated, compared with liquid culture fermentation, SSF of *C. militaris* takes longer to produce, has a low space utilization ration and has a complicated operation process. Many attempts have been made to enhance COR production. However, these attempts to increase COR content are performed in different strains. Thus, more research should be done to verify the effect of these attempts. In the future, different factors affecting the production of COR should be taken into consideration, and various factors affecting the production of COR during *C. militaris* growth should be systematically optimized.

As the genomic sequence of *C. militaris* has been completed, genetic modification technology has been established which greatly promotes the biosynthesis and regulation mechanism of COR. Based on the genomic sequence, the gene cluster Cns1-4 responsible for COR biosynthesis has been identified. Separation and purification methods are also very important for the production of COR. According to the different methods of *C. militaris* fermentation, different separation and purification strategies should be selected and usually many methods are combined to form a high-efficiency and low-cost separation process.

In summary, as COR has great application value in medicine and health-care products, a lot of research work has been done on COR synthesis. Although its biosynthetic pathway has been illuminated, the regulatory mechanism is not clear. With the genomic sequence of *C. militaris* and the development of genetic manipulation technology of *C. militaris*, the regulation mechanism of COR biosynthesis is expected to be gradually revealed. Therefore, it is a possible way to improve the yield of COR by genetic modification of key genes affecting COR biosynthesis. Genetic modification, optimized fermentation conditions and suitable separation and purification methods will greatly promote COR production in the further. Besides, heterologous expression of COR biosynthetic genes using synthetic biology method is also a development direction of COR production in the future, and this has been successful in *S. cerevisiae*, although the yield is relatively low.

## Figures and Tables

**Figure 1 bioengineering-09-00069-f001:**
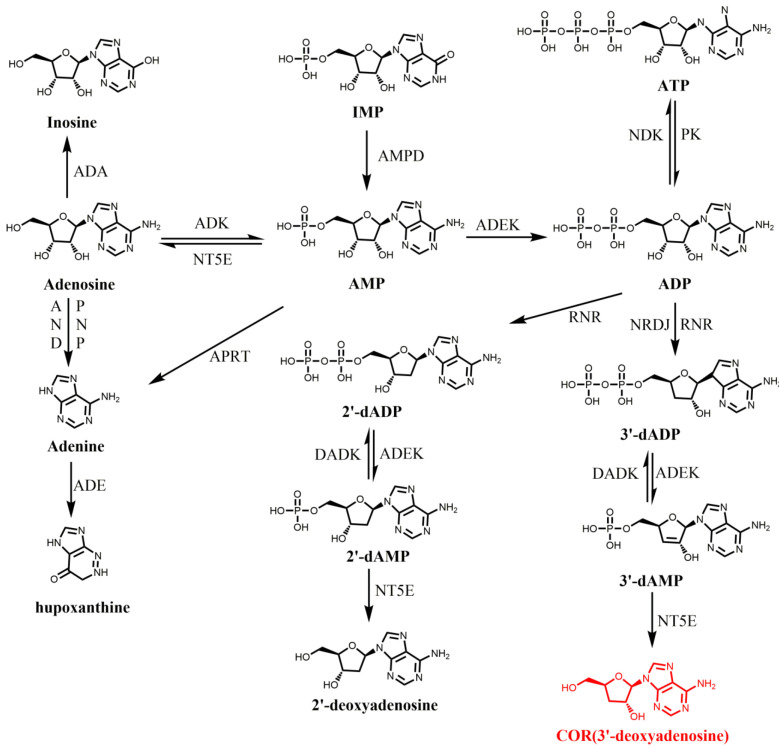
The adenosine metabolism pathway and possible COR biosynthesis pathway [60,61,62]. Abbreviations: IMP: inosine monophosphate; AMP: adenosine monophosphate; ADA, adenosine deaminase; ADE, adenine deaminase; ADK: adenosine kinase; ADEK, adenylate kinase; ADN, adenosine nucleosidase; ADK, adenosine kinase; DADK, deoxyadenylate kinase; NDK, nucleoside-diphosphate kinase; AMPD, AMP deaminase; dAMP: deoxyadenosine monophosphate; ADP: adenosine diphosphate; dADP: deoxyadenosine diphosphate; RNR: ribonucleotide reductases; APRT, adenine phosphoribosyltransferase; NT5E, 5′-nucleotidase; PNP, purine nucleoside phosphorylase; RNR, ribonucleotide triphosphate reductase.

**Figure 2 bioengineering-09-00069-f002:**
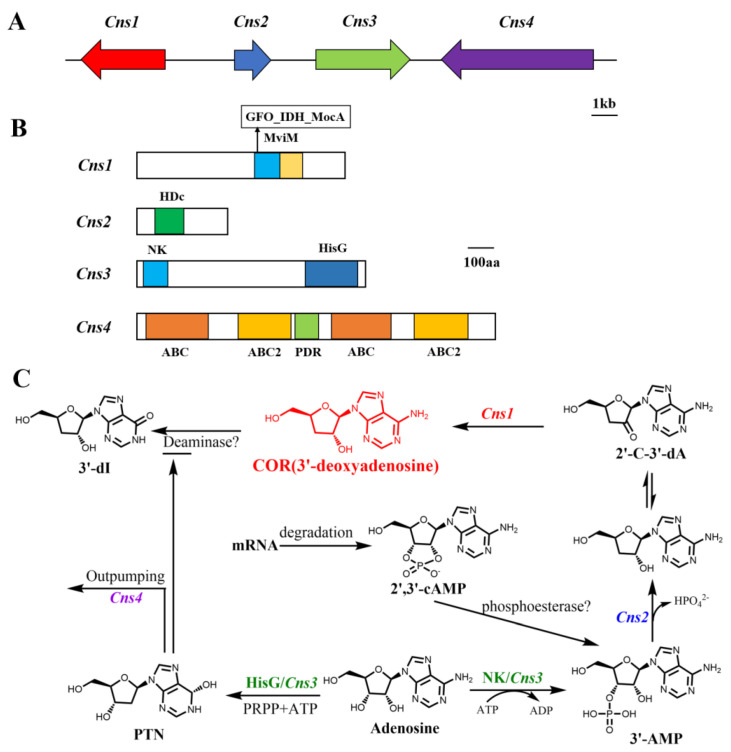
Delineation of COR biosynthetic gene cluster and the COR biosynthesis pathway [63]. (**A**) The gene cluster for synthesize cordycepin *C. militaris*. (**B**) Schematic structure of the proteins. (**C**) Delineation of the biosynthetic pathway of cordycepin. Abbreviations: PRPP, phosphoribosyl pyrophosphate; 2′-C-3′-dA: 2′-carbonyl-3′-deoxyadenosine; 2′,3′-cAMP: 2′,3′-cyclic monophosphate; NK: an N-terminal nucleoside kinase; HisG: a C-terminal HisG family of ATP phosphoribosyltransferases; PTN: pentostatin; 3′-dI: 3′-deoxyinosine.

**Figure 3 bioengineering-09-00069-f003:**
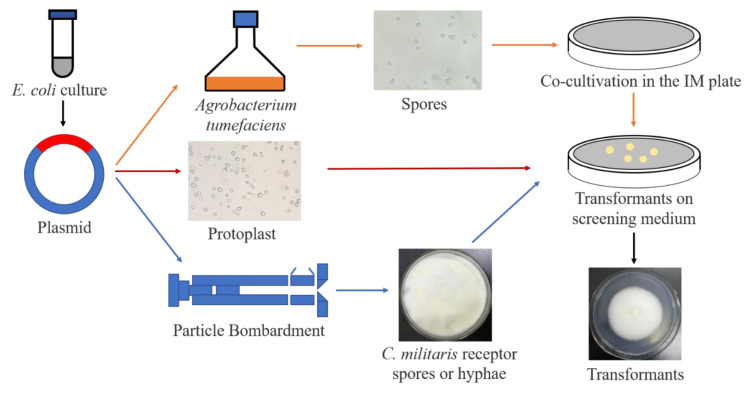
Genetic modification methods of *C. militaris*. Constructed plasmid with target gene was extracted from *E. coli* and transformed by ATMT, PMT and PB methods. For the ATMT method, the plasmid was transferred into *Agrobacterium*, followed by co-cultivation with *C. militaris* spores. For the PMT method, the plasmid (including split marker DNA) was transferred into protoplast mediated by PEG (polyethylene glycol). For the PE method, the plasmid was coated with gold powder, and the broken mycelium suspension of *C. militaris* plated in medium was used as the receptor material for transformation. Finally, the transformant was screened in the corresponding medium for all methods. The transformants were further confirmed by polymerase chain reaction (PCR) or the expression of a reporter gene.

**Figure 4 bioengineering-09-00069-f004:**
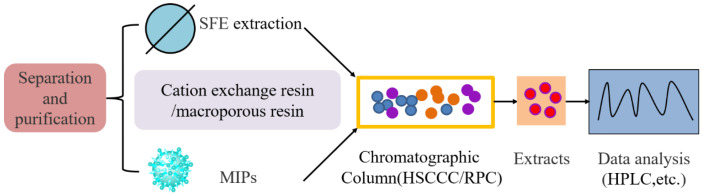
Schematic diagram of COR separation, purification and detection. SFE and MIPs techniques were used to separate the fruiting bodies of solid fermentation or the fermentation broth of liquid fermentation. HSCCC/RPC techniques were used to separate and purify the crystallization of the crude samples, and then the COR samples were detected and analyzed by high-performance liquid chromatography (HPLC).

**Table 1 bioengineering-09-00069-t001:** The starting materials and yield of chemical synthesis of cordycepin (COR).

Starting Material	Final Product	Yield	(Dis)Advantages	Refs
**1** 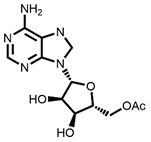	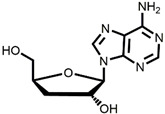 **Cordycepin, COR**	14%	Available raw materials and simple procedure with low yield	[48]
**2** 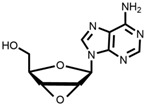		90%	High yield but expensive raw materials	[49]
**3** 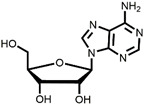		56% (90% per step)	High residual tin	[50,51]
		20%	Mild and controllable reaction conditions, high purity of COR; low yield and high cost	[52]
		13.5%	Simple route with low yield	[53]
		36%	Acceptable total product yield and commercial availability of all starting materials	
**4** 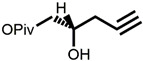		14%	Expensive chemical reagents and low yield	[54]
**5** 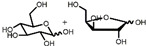		37%; 40%	Cheap raw materials and appropriate yield	[55]

**Table 2 bioengineering-09-00069-t002:** Production of COR under different liquid fermentation conditions.

Strains	Method	COR (g/L)	Increase Rate	Refs
KSP8	Brown rice medium Silk worm pupae medium	6.63 8.10	116% 164%	[68]
G81-3	Medium composition optimization	6.84	179%	[69]
	Adenosine addition	8.60	251%	[70]
NBRC 9787	Adenine and glycine	2.50	400%	[59]
*C. militaris* (Hubei)	Carbon and nitrogen sources optimization	0.35	40%	[71]
CCRC 32219	Two-stage shaking-static culture	2.21	112%	[72]
*C. militaris* (Hubei)	FeSO_4_ addition	0.59	70%	[74]
*C. militaris* BCRC34380	Porcine liver extracts	2.45	390%	[9]
	Combined with blue light LED irradiation	3.48	700%	[9]
*P. hepiali* ZJB18001	Mutant from ^60^Co γ-ray and UV irradiation	0.61	134%	[75]
	Medium composition optimization	0.96	269%	[75]

Note: The increase rate was calculated by the ratio of the increased COR content of different methods to the control. For KSP8, the COR content in parent strains as reference and for *P. hepiali* ZJB18001, the COR content of wildtype was set as reference. The KSP8 strain was obtained by gene recombination of matting; and *P. hepiali* ZJB18001 was obtained by ^60^Co γ-ray and ultraviolet irradiation. The corresponding medium composition optimization methods are described in the main text.

**Table 3 bioengineering-09-00069-t003:** Production of COR under different solid-state fermentation (SSF) conditions.

Strains	Method	COR (mg/g)	Increase Rate	Refs
*C. militaris* (Taiwan)	Ultraviolet B (UVB) irradiation	0.08	10%	[76]
*C. militaris* 34164	Wheat medium	565.20	Control	[77]
	Oat medium	638.85	13%	
	Rice medium	814.60	44%	
*C. militaris* CGMCC2459	Optimization medium composition	9.17	63%	[78]
*C. militaris* 101	Fluorescent lamps	2.89	Control	[80]
	12 h/day illumination	3.97	37%	
	LED wavelengths combinations (3R:3B)	30	938%	

Note: The increase rate was calculated by the ratio of increased COR content of different methods to the control. *C. militaris* 34,164 in wheat medium was set as control, and *C. militaris* 101 under fluorescent lamps was set as control.

**Table 4 bioengineering-09-00069-t004:** Different genetic transformation methods of *C. militaris*.

Strains	Method	Selection Marker	Transformation Efficiency	Refs
JM4	*Agrobacterium tumefaciens*-mediated transformation (ATMT)	Hygromycin B	30–600 cfu/1 × 10^5^ spores	[82]
HF 374-1, HF 438, and CM 001-5	Protoplast-mediated transformation (PMT)	Benomyl	7 cfu/μg	[83]
CM01	Particle Bombardment	Basta	0.4 cfu/µg	[84]
CM10	Split-Marker	Basta	4.53 cfu/µg	[85]
CM10	CRISPR-Cas9	Basta	1.7 cfu/µg	[87]

Note: For ATMT, the transformant is obtained by co-cultivation *Agrobacterium*-*tumefaciens* with spores. Thus, the transformation efficiency is calculated based on the concentration of spores. For other methods, the transformation efficiency is calculated based on the concentration of plasmid.

**Table 5 bioengineering-09-00069-t005:** COR contents in different genetic modified *C. militaris* strains.

Strains	Genetic Modification	COR (mg/g)	Increase Rate	Refs
CGMCC 5.2191	Albino mutant strain	6.700	117%	[89]
*C. militaris* 40 (CGMCC 3.16322)	*CmWC-1c* deletion	0.005	−58%	[90]
*C. militaris*	The RNR subunit RNRM overexpression	3.750	50%	[91]
	The RNR subunit RNRL overexpression	2.600	4%	[91]
ZA10-C4	Hybrid from ZGMM and CM 17 strains	357.166	136–209%	[92]
*C. militaris* CM10	Overexpression of CmTf1 and CmTf2	0.099	100%	[93]
*S. cerevisiae*	Overexpression Cns1 and Cns2	0.021	*S. cerevisiae* can’t synthesis COR	[63]

Note: The increase rate was calculated by the ratio of increased COR content of different genetic modified strains to the control. As in the ∆*CmWC*-1 mutant, COR content was decreased, the increased rate was negative and it was calculated by using the COR content in complemented strain as reference and *S. cerevisiae* cannot synthesis COR.

**Table 6 bioengineering-09-00069-t006:** Methods for separation and purification of COR.

Strains	Method	COR (mg/g)	Purity	Refs
*C. kyushuensis*	supercritical fluid extraction (SFE) and high-speed counter-current chromatography (HSCCC)	22.3	98.5%	[94]
*C. militaris* (Link)	HSCCC and cation-exchange resin	299.7	98.9%	[95]
*C. militaris* (Jiangsu)	Macroporous resin and HSCCC	31.2	98.5%	[96]
*C. militaris*	Silica gel column chromatography and reversed-phase (RPC)	0.4	99%	[97]
*C. militaris* (Shanghai)	Surface imprinting technology	95.4	98%	[98]

## Data Availability

Data is contained within the article.

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
