# Peer review of "Research Progress on Cordycepin Synthesis and Methods for Enhancement of Cordycepin Production in Cordyceps militaris"

_bioengineering, 2022, doi:10.3390/bioengineering9020069_

Round 1
Reviewer 1 Report
This manuscript sorts out the research on COR synthesis and COR mass production and its improvement in yield. The context discusses a fascinating topic, containing several interesting experimental pieces of evidence. But it is a pity that the rigor and accuracy of the data displayed in each table must be clarified before it is suitable for publication.
Main comment
- The data unit in Table 2 needs further confirmation. The COR content of KSP8 is 6.63 mg/g (ref. 58, Figure 5). The author should carefully clarify the formula of the culture medium inoculated in the original paper and the conversion method to g/L in this manuscript. In addition, all data appearing in the tables should be reconfirmed.
- The data in Table 3 is improperly quoted. The 11.15mg/g presented in Table 3 (ref. 65) is not COR. In addition, there is no data of 814.60 mg/g in the original manuscript.
- The "Increase rate" calculation method should be explained at the bottom of all tables (footnote). The author calculated the data of C. militaris 34164 (ref. 66) based on Wheat medium (565.20 mg/g). Appropriate signs are required to guide readers to read. In addition, how are the 32.73% and 44.12% presented by C. militaris (ref. 65) calculated? Table 3.
- At the end of the article, there is a lack of a conclusion that can explain the value of this manuscript.
- The rules for units and significant figures must be consistent. The difference between "7 colonies/μg" and "0.4 cfu/μg" (Table 4)? 13.24% and 64.7% (Table 4) and 157.46%, -212.5% (Table 5). The Significant figures are inconsistent.
- Line 403. "IMPs"?
- What does it mean? "Start from nothing" (Table 5)
- In Table 1, "90% per step" is misplaced (Reference No. 40, 41).
- Extracts or Extractor? (Figure 3)
Author Response
Response to reviewer 1:
This manuscript sorts out the research on COR synthesis and COR mass production and its improvement in yield. The context discusses a fascinating topic, containing several interesting experimental pieces of evidence. But it is a pity that the rigor and accuracy of the data displayed in each table must be clarified before it is suitable for publication.
Main comment:
- The data unit in Table 2 needs further confirmation. The COR content of KSP8 is 6.63 mg/g (ref. 58, Figure 5). The author should carefully clarify the formula of the culture medium inoculated in the original paper and the conversion method to g/L in this manuscript. In addition, all data appearing in the tables should be reconfirmed.
Response: We sincerely appreciate the valuable comments. We are sorry for the unclear description the data in the Table 2. In the fourth part of the paper mainly describes the production of COR by liquid fermentation, so COR content data units are uniformly converted to g/L in the revised manuscript. We checked the data from the original literatures and added the culture medium formula, the following contents in the revised manuscript (Lines: 246-251):
The KSP8 strain COR content increased 116%-164% comparing with average content of COR in 12 the parent strains (3.06 g/L in brown rice medium). The KSP8 strain was inoculated in brown rice (mixing 40g brown rice and 4g silk worm pupae in 64 mL liquid medium) and silk worm pupae (20 g brown rice, 40 g silk worm pupae in 64 mL liquid medium) mediums, respectively; the liquid medium formula was as follows: 20 g sucrose, 20 g peptone, 1 g MgSO4·7H2O, 0.5 g KH2PO4, and 1 L distilled water. By using ion beam irradiation, a C. militaris mutant strain G81-3 was obtained, and the yield of COR reached 6.84 g/L in surface liquid culture (the optimal concentrations of glucose and yeast extract for the mutant strain were 86.2 g/L and 93.8 g/L, respectively), which was 2.79 times than that of the corresponding wild strain (2.45 g/L). The yield of COR increased to 8.6 g/L by using G81-3 strain with 6 g/L adenosine addition, which was 3.51 times than that of the wild strain. The uncertain data in the table 2 were corrected, 60Co γ-ray and ultraviolet irradiation were employed to increase the COR content, and resulted in a high-yield mutant strain P. hepiali ZJB18001 with the COR content of 0.61 g/L, 2.3-fold than that of wild type strain (0.26 g/L) was screened. Furthermore, medium composition optimization (peanut oil 21 g/L, yeast extract 9.8 g/L, Fe2(SO4)3 4.68 mmol/L and adenosine 2.14 g/L) based on Box-Behnken design and response surface methodology facilitated the enhancement of COR yield to 0.96 g/L at 25°C for 5 days in submerged cultivation.
- The data in Table 3 is improperly quoted. The 11.15mg/g presented in Table 3 (ref. 65) is not COR. In addition, there is no data of 814.60 mg/g in the original manuscript.
Response: Thanks very much for your comments. We are sorry for the wrong description here. We read the original paper carefully and corrected for previous improperly quoted. The UVB light irradiated dry samples (buckwheat and embryo rice contained COR 0.08 mg/g) fermented by C. militaris was 1.1-fold higher than that of the irradiated fresh samples fermented (0.07 mg/g). The data of 814.60 mg/g is COR content using C. militaris 34164 in rice medium but not C. militaris (Taiwan) and we have checked the original paper (DOI: 10.1080/19476337.2017.1325406). In order to avoid misunderstanding, we revised the format of table 3 and other tables in the revised manuscript.
- The "Increase rate" calculation method should be explained at the bottom of all tables (footnote). The author calculated the data of militaris 34164 (ref. 66) based on Wheat medium (565.20 mg/g). Appropriate signs are required to guide readers to read. In addition, how are the 32.73% and 44.12% presented by C. militaris (ref. 65) calculated?
Response: Thanks for your suggestion, and we have revised it in Table 3. Besides, we also adjusted data in Table 3, making it more suitable for reading. The calculation method of COR increase rate has been added to the full-text table and explained at the bottom of each table (footnotes have been added). The increase rate is the ratio of increased COR content of different methods to the control (wild type). In addition, C. militaris 34164 based on wheat medium (565.20 mg/g) was set as control to calculate the increase rate of COR in oat and rice medium. All the calculation methods were added at the bottom of tables in the revised manuscript.
- At the end of the article, there is a lack of a conclusion that can explain the value of this manuscript.
Response: Thank you for your valuable comments. Actually, we have the contents of conclusion in the discussion part. For example, the first paragraph of discussion is to summarize and discuss chemical synthesis and biosynthesis of COR, and the second paragraph is to summarize and discuss the method to increase COR content in liquid and SSF fermentation, the third paragraph is to summarize and discuss genetic modification of C. militaris to increase COR content and way of COR separation and purification also included. Therefore, we changed the title of this section as ‘Conclusion and Discussion’. Besides, we added contents perspectives the last paragraph in this section (Lines: 532-542).
- The rules for units and significant figures must be consistent. The difference between "7 colonies/μg" and "0.4 cfu/μg" (Table 4)? 13.24% and 64.7% (Table 4) and 157.46%, -212.5% (Table 5). The Significant figures are inconsistent.
Response: Thank you for your valuable comments. We apologize for the inconsistent of significant figures in the manuscript and we revised the significant figures in the manuscript. And we unified the ‘colonies’ as ‘cfu’ in the table 4. The calculation method has been supplemented in footnotes.
- Line 403. "IMPs"?
Response: Thank you for your comments. We are sorry for the mistake and we have corrected it as ‘MIPs’ in the revised manuscript (Line 478 in the revised manuscript).
- What does it mean? "Start from nothing" (Table 5)
Response: We are sorry for the unclear description here. As S. cerevisiae can't synthesis COR, the increase rate can’t be calculated, we have corrected description as ‘S. cerevisiae can't synthesis COR’ in the revised manuscript (Line 428 in the revised manuscript). And there is also footnote at the bottom of the table to explain it.
- In Table 1, "90% per step" is misplaced (Reference No. 40, 41).
Response: We are sorry for the mistake and the correct place has been corrected in the revised manuscript.
- Extracts or Extractor? (Figure 3)
Response: We are sorry for the mistake and we have corrected it as ‘extracts’ in figure 3 and related descriptions are also revised.
Reviewer 2 Report
Dear Authors
In this manuscript entitled: Research Progress on Cordycepin Synthesis and Methods for enhancement of Cordycepin production in Cordyceps militaris, authors described in a proper way the mechanism of chemical synthesis of cordycepin by using very good figure and table, than the adenosine metabolism pathway and possible biosynthesis pathway. In the third paragraph, authors reviewed the liquid fermentation of cordycepin under different experimental conditions.
Authors described the improvement of cordycepin production by using genetic modified C. militaris fungus.
The discussion part is well designed and the references used supported the hypothesis of the manuscript title.
Comment:
Please add a conclusion/perspectives part
Hence, I strongly recommend the acceptance of this excellent review paper
Best regards
Author Response
Reviewer 2
In this manuscript entitled: Research Progress on Cordycepin Synthesis and Methods for enhancement of Cordycepin production in Cordyceps militaris, authors described in a proper way the mechanism of chemical synthesis of cordycepin by using very good figure and table, than the adenosine metabolism pathway and possible biosynthesis pathway. In the third paragraph, authors reviewed the liquid fermentation of cordycepin under different experimental conditions. Authors described the improvement of cordycepin production by using genetic modified C. militaris fungus. The discussion part is well designed and the references used supported the hypothesis of the manuscript title.
Comment:
- Please add a conclusion/perspectives part
Response: Thanks very much for the comments and suggestions. Actually, the conclusion of this review was included in the discussion part. Therefore, we changed the title of this section as ‘Conclusion and Discussion’. Besides, we added contents perspectives in the last paragraph of this section (Lines: 532-542).
Reviewer 3 Report
The current review on "Research Progress on Cordycepin Synthesis and Methods for Enhancement of Cordycepin Production in Cordyceps militaris " by Li et al, is a good piece of story where authors tried to combine COR synthesis and methods involved in enhancement of COR production. In general, this review has written well and no serious flaws I have found. But at the present format this review is not suitable for the publication in Bioengineering Journal. Therefore, I request the authors to modify this review article based on following suggestions:
- I would like to see a separate paragraph/section for biology activity of COR. Though it is mentioned a bit in introduction part, but a separate section would be much better for this review article.
- Genetic modification of the strain C militaris should be added in the main text separately with figure.
- The future prospect of the COR production with respect to strain modification should be added
Round 2
Reviewer 1 Report
Paper has been improved after recommendation.